# Disruption in structural–functional network repertoire and time-resolved subcortical fronto-temporoparietal connectivity in disorders of consciousness

Rajanikant Panda[1,2], Aurore Thibaut[1,2], Ane Lopez-Gonzalez[3], Anira Escrichs[3], Mohamed Ali Bahri[4], Arjan Hillebrand[5], Gustavo Deco[3,6,7,8], Steven Laureys[1,2,9], Olivia Gosseries[1,2], Jitka Annen[1,2*†], Prejaas Tewarie[5,10*†]

[1]Coma Science Group, GIGA-Consciousness, University of Liège, Liège, Belgium; [2]Centre du Cerveau, University Hospital of Liège, Liège, Belgium; [3]Computational Neuroscience Group, Center for Brain and Cognition, Universitat Pompeu Fabra, Bracelona, Spain; [4]GIGA-Cyclotron Research Centre-In Vivo Imaging, University of Liège, Liège, Belgium; [5]Amsterdam UMC, Vrije Universiteit Amsterdam, Department of Clinical Neurophysiology and MEG Center, Amsterdam Neuroscience, Amsterdam, Netherlands; [6]Institució Catalana de la Recerca i Estudis Avançats (ICREA), Barcelona, Spain; [7]Department of Neuropsychology, Max Planck Institute for Human Cognitive and Brain Sciences, Leipzig, Germany; [8]School of Psychological Sciences, Monash University, Melbourne, Australia; [9]CERVO Research Center, Laval University, Québec, Canada; [10]Sir Peter Mansfield Imaging Centre, School of Physics and Astronomy, University of Nottingham, Nottingham, United Kingdom

*For correspondence:
jitka.annen@uliege.be (JA);
prejaas.tewarie@nottingham.ac.uk (PT)

†These authors contributed equally to this work

**Abstract** Understanding recovery of consciousness and elucidating its underlying mechanism is believed to be crucial in the field of basic neuroscience and medicine. Ideas such as the global neuronal workspace (GNW) and the mesocircuit theory hypothesize that failure of recovery in conscious states coincide with loss of connectivity between subcortical and frontoparietal areas, a loss of the repertoire of functional networks states and metastable brain activation. We adopted a time-resolved functional connectivity framework to explore these ideas and assessed the repertoire of functional network states as a potential marker of consciousness and its potential ability to tell apart patients in the unresponsive wakefulness syndrome (UWS) and minimally conscious state (MCS). In addition, the prediction of these functional network states by underlying hidden spatial patterns in the anatomical network, that is so-called eigenmodes, was supplemented as potential markers. By analysing time-resolved functional connectivity from functional MRI data, we demonstrated a reduction of metastability and functional network repertoire in UWS compared to MCS patients. This was expressed in terms of diminished dwell times and loss of nonstationarity in the default mode network and subcortical fronto-temporoparietal network in UWS compared to MCS patients. We further demonstrated that these findings co-occurred with a loss of dynamic interplay between structural eigenmodes and emerging time-resolved functional connectivity in UWS. These results are, amongst others, in support of the GNW theory and the mesocircuit hypothesis, underpinning the role of time-resolved thalamo-cortical connections and metastability in the recovery of consciousness.

## Editor's evaluation

This is an important paper providing convincing evidence for altered brain dynamics in patients in a minimally conscious state and those with unresponsive wakefulness syndrome relative to healthy control participants. The results indicate reduced metastability and a contracted network repertoire in disorders of consciousness. Overall, the study provides important new information on mechanisms of disorders of consciousness and the functional brain networks involved.

## Introduction

Diagnosis of the level of consciousness after coma due to severe brain injury is a well-known dilemma in the field of neurology and intensive care medicine. Coma after cardiac arrest or after traumatic brain injury may result in sustained altered states of consciousness. These patients with disorders of consciousness (DoC), irrespective of the aetiology, can be grouped into the unresponsive wakefulness syndrome (UWS) (*Laureys et al., 2010*), characterized by the presence of eye-opening and reflexive behaviours, and the minimally conscious state (MCS), characterized by consistent but fluctuant wilful conscious behaviours, such as command following or visual pursuit (*Giacino et al., 2002*; *Schnakers et al., 2010*). Recovery of consciousness is argued to emerge conjointly with restoration of resting-state functional brain networks (*Edlow et al., 2021*), which refers to patterns of neuronal interactions inferred by indirect (e.g. functional magnetic resonance imaging – fMRI) or direct (e.g. electro- and magneto-encephalography [EEG/MEG]) measurements. Analysis of these resting-state networks could potentially help in the diagnosis of patients with DoC and provide insight into the mechanisms that results in absence of recovery of consciousness in UWS.

Various resting-state networks that play an important role in the recovery of consciousness have been identified, among which the default mode network (DMN), fronto-parietal network (FPN), and the salience network are the most important (*Amico et al., 2017*; *Heine et al., 2012*). Recovery of the DMN in combination with recovery of the auditory network could for instance discriminate between MCS and UWS with a very high accuracy (~85%) (*Demertzi et al., 2015*). The mechanism of resting-state network restoration in DoC is yet unknown, however, thalamic activity and especially thalamo-cortical connectivity may be a driving force behind restorations of cortical network function that sustains conscious states (*Fridman et al., 2014*; *Laureys et al., 2000*). Previous work on resting-state networks in DoC have mainly focused on the '*static*' picture of functional connectivity (*Demertzi et al., 2013*; *Edlow et al., 2021*; *Giacino et al., 2014*; *Heine et al., 2012*), that is connections are assessed over the entire duration of the (fMRI) recording and fluctuations in connectivity over time are ignored. However, the underlying dynamics of connectivity seem relevant for consciousness (*Barttfeld et al., 2014*; *Luppi et al., 2019*) and a static description may therefore be inadequate to provide mechanistic insight into failure of recovery of consciousness in DoC (*Demertzi et al., 2019*).

The analysis of dynamic or time-resolved functional connectivity, as well as the relationship between the underlying anatomical connections and emergent time-resolved functional connectivity (*Avena-Koenigsberger et al., 2018*; *Suárez et al., 2020*), may be clinically relevant in patients with DoC. Previous studies have already explored the role of time-resolved functional connectivity in DoC (*Del Pozo et al., 2021*; *Golkowski et al., 2021*; *Sanz Perl et al., 2021*). A recent study demonstrated that network states with long-distance connections occurred less frequently over time in MCS compared to UWS patients (*Demertzi et al., 2019*), emphasizing disintegration of interactions across the cortex in unconscious states. However, network states reminiscent of the well-known resting-state networks were not retrieved. Cao et al. used two methods to extract time-varying networks, that is independent component analysis and hidden Markov modelling, and revealed clinically relevant differences in network state durations between patients with DoC patients and healthy subjects (*Cao et al., 2019*), while lacking comparative analysis between patients in MCS and in UWS. In another fMRI study, the authors focused on the posterior cingulate area and the DMN using a spatiotemporal point process analysis and demonstrated decreased occurrence of DMN-like patterns in UWS. Dynamic connectivity analysis has also recently been applied to EEG data, revealing a loss of network integration and increased network segregation in DoC patients (*Rizkallah et al., 2019*). Spatiotemporal properties of networks have been explored using whole brain modelling, which shows reduction in stability, heterogeneity, and information flow in loss of consciousness (*Escrichs et al., 2021*; *López-González et al., 2021*; *Panda et al., 2021*). Despite the importance of the previously published work, the role of the well-known resting-state networks and especially thalamo-cortical functional connections

(*Monti et al., 2015*) within the context of time-resolved connectivity and DoC has so far not been fully explored.

Another important aspect in the context of the emergence or restoration of resting-state networks is the underlying structural network, as anatomical connectivity patterns influence the repertoire of possible functional network states (*Deco et al., 2013*). It is widely assumed that switching between functional network states is achieved by so-called metastability in the brain (*Deco and Kringelbach, 2016*), that is winnerless competitive dynamics. A promising and robust approach to analyse the relationship between structural and functional network states is the so-called eigenmode approach (*Atasoy et al., 2016*; *Robinson et al., 2016*; *Tewarie et al., 2020*). With this approach, spatial harmonic components or eigenmodes are extracted from the anatomical network. These eigenmodes can be considered as patterns of 'hidden connectivity' that come to expression at the level of functional networks. It has been postulated that eigenmodes form elementary building blocks for spatiotemporal brain dynamics (*Aqil et al., 2021*). There is evidence that the well-known resting-state networks can be explained by the activation of a small set of eigenmodes (*Atasoy et al., 2018*). It can be hypothesized that switching between functional network states, as can be observed in the metastable brain, is accompanied by fluctuations in the expression of eigenmodes *Preti and Van De Ville, 2019*; therefore, a potential loss of metastability in DoC could co-occur with loss of modulations in eigenmode expression (*Barttfeld et al., 2014*).

In this context, the aim of the current study was fourfold. First, we tested whether loss of metastability and resting-state network activity, derived from time-resolved estimates of functional connectivity, could differentiate between MCS and UWS patients, with a potential extraction of a spatiotemporal thalamo-cortical network state. Second, we analysed whether time-resolved connectivity could be explained by modulations in expression of eigenmodes in DoC, and third, whether potential differences in eigenmode expression in DoC patients would co-occur with a loss of metastability. Finally, we conceptually link the findings of altered spatiotemporal dynamics underlying activity observed in the brain of DoC patients with several consciousness theories to increase our understanding of the mechanism behind pathological states of unconsciousness.

## Results

We included 34 healthy control subjects (HC, 39 [mean] ± 14 years [standard deviation], 20 males), 30 MCS (41 ± 13 years, 21 males) and 14 UWS patients (48 ± 16 years, 7 males). There was no difference in patients with MCS and UWS in terms of age (p > 0.05), gender (p > 0.05), time since injury (*p* > 0.05), and aetiology (p > 0.05). There was also no difference in age (p > 0.05) and gender (p > 0.05) between HC and DoC patients. Further details about the patient population are described in the methods and *Supplementary file 1*.

### Metastability and time-resolved functional connectivity in patients with DoC

Time-resolved or dynamic connectivity for all subjects was extracted from the phase information of the fMRI data. We quantified a proxy measure for metastability defined as the standard deviation of the overall phase behaviour over time (i.e. the Kuramoto order parameter). This was followed by the extraction of spatiotemporal patterns using non-negative tensor factorization (NNTF) from phase connectivity data, corresponding to resting-state networks or network states (see *Figure 1*). Well-known resting-state networks as well as a residual component were used as initial conditions for spatial connectivity patterns for all network states to allow for stable convergence of the algorithm (i.e. DMN, FPN, visual network, sensorimotor network, salience network, and subcortical network; *Finn et al., 2015*). However, the NNTF algorithm allowed the spatial patterns of these network states to change in order to maximize the explained variance of the data. Temporal statistics from the network states were derived for every network state in terms of excursions from the median proxy for nonstationarity (*Zalesky et al., 2014*) and state duration (i.e. dwell time).

A reduction of metastability was found in DoC patients compared to HCs (*Figure 2A*). Lower metastability was observed in UWS patients in comparison to MCS patients (*Figure 2A*). Reduced metastability is expected to occur with loss of switching between resting-state networks and potentially with dwelling within a more limited subset of resting-state networks in DoC. The output of the

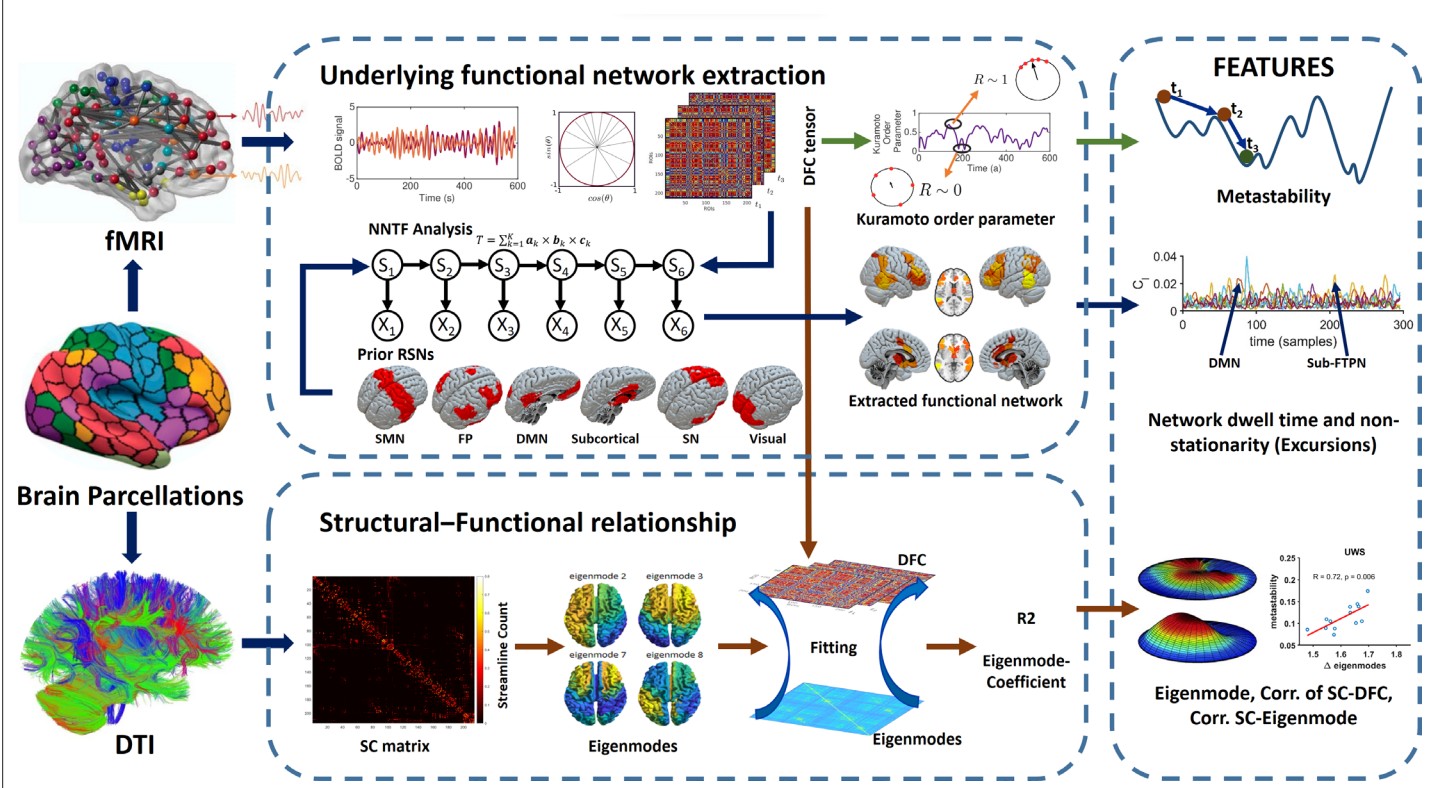

**Figure 1.** Overview of the analysis pipeline. We used the same Shen parcellation for diffusion-weighted MRI (DWI) and functional magnetic resonance imaging (fMRI) data. Time-resolved functional connectivity was estimated using a metric for phase connectivity. A proxy measure for metastability was derived from the phase information. Time-resolved networks were subsequently extracted from the concatenated data from all subjects using non-negative tensor factorization (NNTF). Dwell times and nonstationarity (excursions from the median) were retrieved for each spatial pattern of functional connectivity (six resting-state networks and one 'residual' network). At the same time, time-resolved connectivity was predicted on a sample-by-sample basis based on a linear combination of eigenmodes (hidden patterns in the anatomical network). Measures were used for classification and feature ranking.

NNTF algorithm resulted in spatial topographies of some of the well-known resting-state networks, that is the DMN, a separate posterior DMN around the precuneus, the visual network, the salience network, the FPN, and a network consisting of subcortical fronto-temporoparietal regions (Sub-FTPN) (*Figure 2I–N*, *Supplementary file 1B*). Note that these network states were not identical to the initial conditions, for example the subcortical network that was provided as initial condition to NNTF was incorporated with the frontoparietal network (*Figure 2N*) by the NNTF algorithm. This modulated subcortical fronto-temporoparietal network consists of the following brain regions: bilateral thalamus, caudate, right putamen, bilateral anterior and middle cingulate, inferior and middle frontal areas, supplementary motor cortex, middle and inferior temporal gyrus, right superior temporal, bilateral inferior parietal, and supramarginal gyrus. At the same time, the sensorimotor network that was provided as initial condition disappeared as state. Excursions from the median were lower for most networks (DMN, visual, Salience, Posterior DMN, and Sub-FTPN) in DoC compared to HC (*Figure 2C–F, H*). Significant loss of nonstationarity was also found in UWS compared to MCS for the DMN, FPN, and Sub-FTPN (*Figure 2C, G, H*). The NNTF also yielded a residual state, with a lack of spatial structure, accounting for the variance of connectivity data not explained by the resting-state networks. The residual component had longer dwell times for the decreasing levels of consciousness (*Figure 2B*). In addition, there were lower dwell times in DoC patients for a specific set of resting-state networks (Salience, Posterior DMN, FPN, and Sub-FTPN), and dwell time was shorter in UWS patients compared to MCS patients only in the Sub-FTPN network (see results in *Figure 2—figure supplement 1*). These findings of very short dwell times in the posterior DMN, FPN, and Sub-FPTN and long dwell time in the residual network can be considered as a contraction of the functional network repertoire in DoC, which is in agreement with a loss in metastability in these patients.

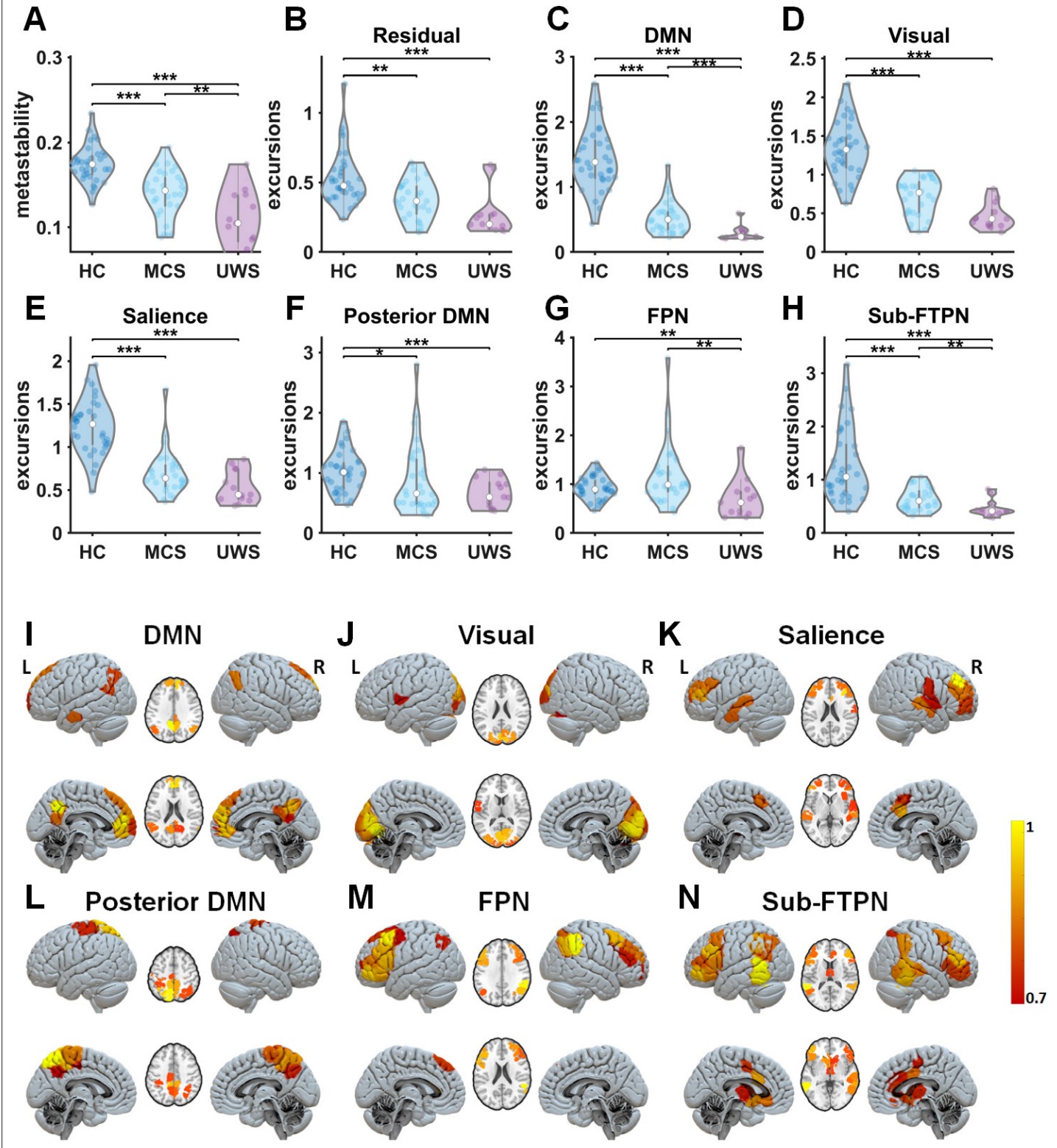

**Figure 2.** Metastability and time-resolved functional networks in disorders of consciousness (DoC). (**A**) Metastability for all groups: healthy controls (HC), minimally conscious state (MCS), and unresponsive wakefulness state (UWS) patients. (**B–H**) Distributions of nonstationarity (excursions from the median) for the residuals and time-resolved networks. (**I–N**) Spatial patterns of time-resolved output networks. Abbreviations: DMN, default mode network; FPN, frontoparietal network; Sub-FTPN, subcortical fronto-temporoparietal network. *, ** and *** denote p < 0.05, p < 0.01 and p < 0.001, respectively (Mann–Whitney *U* tests). The colourbar indicates the strength of that specific area to the overall spatial pattern. Note, the Sub-FTPN appears as

*Figure 2 continued on next page*

*Figure 2 continued*

modified network from the initially assigned subcortical network. This Sub-FTPN consists of the following brain regions: bilateral thalamus, caudate, right putamen, bilateral anterior and middle cingulate, inferior and middle frontal areas, supplementary motor cortex, middle and inferior temporal gyrus, right superior temporal, bilateral inferior parietal, and supramarginal gyrus.

The online version of this article includes the following figure supplement(s) for figure 2:

**Figure supplement 1.** Dwell time of time-resolved functional networks in disorders of consciousness (DoC).

## Relationship between structural eigenmodes and time-resolved functional connectivity in DoC

As resting-state network activity can be explained by the activation of structural eigenmodes, we next analyse the role of fluctuations in eigenmode expression over time. In order to put our findings into context, we first analysed the relationship between static functional networks and structural networks, using the Pearson correlation between static functional connectivity and structural connectivity (SC) for the different groups (*Figure 3A*). These results show that functional connectivity in DoC patients

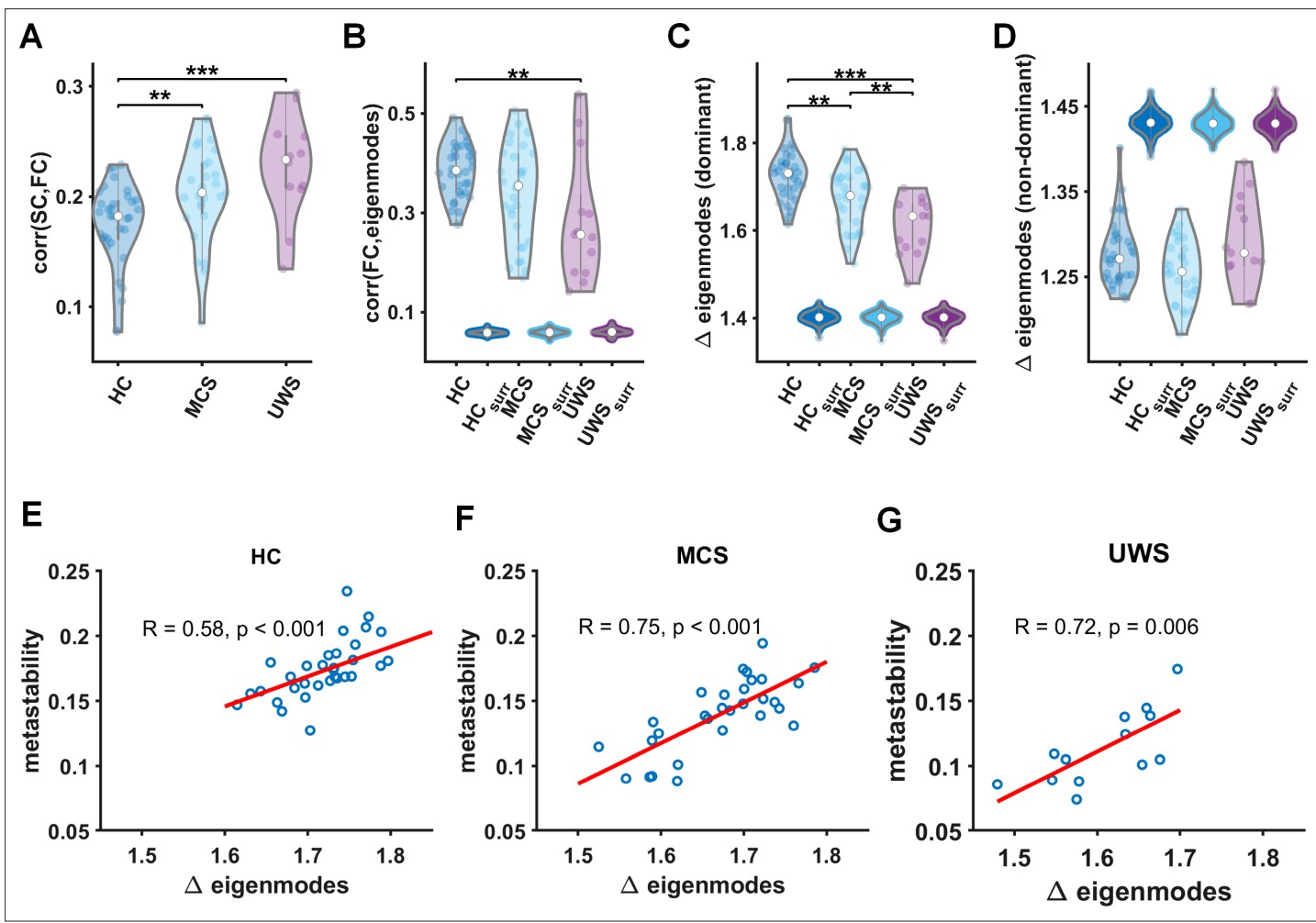

**Figure 3.** Relationship between time-resolved connectivity and eigenmodes. (**A**) The prediction of static functional connectivity based on structural connectivity for all three groups (healthy control [HC], minimally conscious state [MCS], and unresponsive wakefulness syndrome [UWS]) in terms of the Pearson correlation coefficient. (**B**) The prediction of time-resolved functional connectivity based on eigenmodes. These distributions of eigenmode predictions are accompanied with predictions based on surrogate data. We further illustrate the level of fluctuations in eigenmode expression for all three groups (HC, MCS, and UWS) for dominant (reflecting network integration, **C**) and non-dominant eigenmodes (reflecting increasing network segregation, **D**) accompanied with results for surrogate data, ** and *** denote $p < 0.01$ and $p < 0.001$, respectively (Mann–Whitney $U$ tests). (**E–G**) Metastability is strongly correlated to modulations in eigenmode expression within every group.

shows more correspondence with the underlying SC as compared to HCs, as the relationship between structural and functional connectivity was stronger for decreasing levels of consciousness (*Figure 3A*).

We next obtained the eigenmodes from the SC by extracting the eigenvectors of the graph Laplacian. These eigenmodes can be regarded as distinct spatial harmonics within the SC, where the first eigenmodes correspond to patterns with low spatial frequency and subsequent eigenmodes contain patterns with increasingly higher spatial frequencies. Given their spatial configuration, consecutive eigenmodes can be associated with increasing levels of segregation while the first eigenmodes can be linked with network integration. For every time point we predicted the extent to which phase connectivity could be explained by a weighted combination of the eigenmodes (*Tewarie et al., 2020*). Since phase connectivity can evolve over time, the weighting coefficients for the eigenmodes can modulate as well, resulting in fluctuations in the strength of the expressions of eigenmodes over time. For every eigenmode, we could then quantify the modulation strength (i.e. how much the eigenmode expression varied over time). In addition to the weighting coefficients, we also obtained the goodness-of-fit for the predictions of time-resolved functional connectivity.

The goodness-of-fit for the eigenmode predictions is displayed in *Figure 3B*, where we show the average correlation between eigenmode predicted FC and empirical FC for the three groups. Results show better predictions for HC and MCS compared to predictions for static FC (median and interquartile range of correlations HC static 0.18 ± 0.04, HC eigenmode 0.39 ± 0.09, $Z = -7.1$, p < 0.001, MCS static 0.2 ± 0.05, MCS eigenmode 0.35 ± 0.18, $Z = -4.8$, p < 0.001). In order to test whether these eigenmode predictions of time-varying connectivity could have been obtained by chance, we redid our analysis using surrogate BOLD data (see methods '*Analysis steps*'). Results show that eigenmode predictions for time-resolved connectivity from surrogate data performed significantly worse compared to genuine empirical data (for all comparisons with surrogate data p < 0.001; *Figure 3B*). We did not test whether the contribution of individual eigenmodes differed between groups since this would come with a serious multiple comparisons problem.

Instead, since SC appeared to correlate stronger with static FC in DoC compared to HC, we expected that eigenmode coefficients in DoC patients would hardly change over time, highlighting the observation of a 'fixed' structural–functional network relationship in DoC patients. To analyse this lack of change in the structural–functional network relationship over time in DoC patients, we quantified the modulation strength of the weighting coefficients over time (see methods '*Analysis steps*'). We performed this analysis separately for the dominant (1st to 107th eigenmode, first half) and non-dominant eigenmodes (108th to 214th eigenmode, second half). Results for the dominant eigenmodes show a clear reduction in modulation of the eigenmode weighting in DoC patients compared to HCs (*Figure 3C*), with a significantly lower modulation of eigenmode expression in UWS compared to MCS patients. This result could not be explained by chance, since the same results could not be obtained from surrogate data (*Figure 3C*). Note that no between group difference was obtained for non-dominant eigenmodes (*Figure 3D*).

We have so far shown a reduction in the modulation strength of eigenmode expressions in DoC patients compared to HC subjects, as well as a loss of metastability in DoC patients and dwelling of the brain in fewer network states in DoC patients. This poses the question of whether these two observations are related. In *Figure 3E–G*, we show that metastability is strongly correlated with modulations in eigenmode expression for the three groups. This underscores the notion that loss of dynamic modulations in functional network patterns due to a loss of metastability could indeed be related to a reduced modulation of eigenmode expression.

## Discussion

Differentiation between MCS and UWS is key for adequate diagnosis and prognosis in DoC patients as it is connected to medical-ethical end of life decisions. Use of imaging characteristics allows testing of hypotheses about causes of delayed, or failure of, recovery of consciousness. Here, we used state-of-the-art techniques to quantify time-varying functional connectivity, metastability, and the relationship between the underlying anatomical network and time-resolved functional connections. We demonstrated that these advanced techniques were sensitive to detect clinically relevant differences for the diagnosis of MCS and UWS patients. More specifically, we first demonstrated that UWS patients showed reduced metastability, and spend less time in states outside the natural equilibrium state that would favour cerebral processing in a cooperative and coordinated manner to

support consciousness. This is accompanied by shorter state durations that the brain spends in the subcortical fronto-temporoparietal configuration in UWS. A loss of nonstationarity was observed in several resting-state networks (i.e. DMN, frontoparietal, and subcortical fronto-temporoparietal) in UWS compared to MCS patients. Finally, we showed that functional brain networks are more 'fixed' to the underlying anatomical connections and are less subject to spatial reconfigurations over time in UWS compared to MCS patients. The extent to which these spatial reconfigurations occurred (i.e. expressed as modulations in eigenmode expression) correlated strongly to metastability.

Our results are in agreement with several hypothesis and theories for the emergence of consciousness, of which most share the importance of thalamo-cortical connectivity for consciousness (*Blumenfeld, 2021*; *Dehaene et al., 2011*; *Schiff, 2010*). The mesocircuit hypothesis states that deafferentation between the frontal cortex and subcortical regions is crucial in explaining the failure of recovery of consciousness (*Schiff, 2010*). One of the most novel findings in the current work is the generation of the subcortical fronto-temporoparietal network. Although subcortical connections were, among others, used as initial conditions for the decomposition of the time-varying functional connectivity patterns into resting-state networks, incorporation with fronto-temporoparietal connections emerged from the data-driven NNTF algorithm. Another observation confirmed that this NNTF approach extracted DoC-relevant networks, namely that the sensorimotor network disappeared after optimization of spatial network patterns. This latter result is in line with the fact that somatosensory cortices are not directly involved in the emergence of consciousness, based on current theories (*Naccache, 2018*). In addition, we found that the subcortical fronto-temporoparietal network showed shorter dwell times in DoC patients compared to HC subjects, with even shorter state durations in UWS compared to MCS patients. Finally, this network also demonstrated a loss of nonstationarity in UWS compared to MCS patients. However, it should be noted that the subcortical fronto-temporoparietal network was not the only network with loss of time-resolved network characteristics; other resting-state networks also showed loss of nonstationarity, such as the DMN and FPN. However, a combination of shorter dwell times and loss of nonstationarity was only found for the subcortical fronto-temporoparietal network.

The mesocircuit hypothesis postulates that increased subcortical (specifically thalamic nuclei) inhibition and reduced excitatory output from the thalamus to the cortex (frontal, parietal, temporal, and occipital) leads to weak neural activity and hampers consciousness in DoC patients (*Fridman et al., 2014*; *Giacino et al., 2014*). Although our method does not allow to assess excitatory and inhibitory activity, it may be tempting to interpret that the subcortical fronto-temporoparietal network may play a crucial role in orchestrating global network interactions and dwell times. Hence, this subnetwork may be instrumental for the observed loss of nonstationarity in the other subnetworks. Although the importance of functional connections between the thalamus and frontal cortex has been emphasized by the mesocircuit hypothesis, and shown to relate to consciousness in hypothesis-driven functional (e.g., *Crone et al., 2018*; *Fridman et al., 2014*; *Monti et al., 2015*) and structural (e.g., *Annen et al., 2016*; *Weng et al., 2017*) neuroimaging studies, this is the first demonstration of the ability of a (semi) data-driven approach to identify this sub-network in the context of time-resolved functional connectivity. Most previous data-driven approaches have been unable to extract such a network (*Cao et al., 2019*; *Demertzi et al., 2019*; *Rizkallah et al., 2019*). Our findings also support the global neuronal workspace (GNW) theory (*Dehaene et al., 2011*), which emphasizes the importance of long-distance, feedforward and recurrent functional connections, large-scale reverberant networks, and metastable brain states in the emergence and recovery of consciousness (*Mashour et al., 2020*). Specifically, the GNW postulates that conscious perception should rely on two neural circuits. First, the feedforward circuit encompasses the visual and sensory areas and propagates sensory input to higher order integration areas. Second, in order for sensory input to be accessible to consciousness, recurrent activity in the circuit is required, responsible for ignition in a wide range of high-order functional areas including visual, parieto-temporal, thalamic, and frontal areas (*Aru et al., 2020*; *Baars, 2002*; *Dehaene and Changeux, 2011*; *Mashour et al., 2020*). The subcortical fronto-temporoparietal network and fronto-parietal network obtained in a data-driven fashion conjugates with the latter recurrent circuit proposed to be essential for consciousness by the GNW recurrent circuit. The breakdown of spatio-temporal coordination in the subcortical fronto-temporoparietal network and frontoparietal network was associated to reduced states of consciousness. So far, the importance of metastability has mainly been addressed in the context of recovery of consciousness from anaesthesia (*Cavanna et al., 2018*). Here, we underscore this finding and demonstrate that a reduction of subcortical–cortical spatiotemporal

functional organization into metastable states can even differentiate between UWS and MCS patients aligning with key factors for consciousness as proposed by GNW and mesocircuit hypothesis.

Despite the overlapping cortical circuits important for the mesocircuit model and GNW, differences between the two hypotheses exist. While the occipital cortex is considered part of the same mesocircuit, its role is mostly for feedforward propagation in the GNW. We found a separate visual network that conceptually overlaps more with the feedforward network described by the GNW. In line with GNW predictions, the temporal dynamics in this feedforward network did not distinguish different states of consciousness (*Dehaene and Changeux, 2011*; *Mashour et al., 2020*). On the other hand, the involvement of subcortical areas in the subcortical fronto-temporoparietal network is specifically predicted by the mesocircuit model (*Fridman et al., 2014*; *Giacino et al., 2014*). Last, the two theories highlight different aspects of brain activity required for consciousness. The GNW proposes specific effective connectivity patterns required for stimuli to reach consciousness. The mesocircuit model, on the other hand, proposes that balanced excitatory and inhibitory activity within the network is a prerequisite for conscious states to be supported. Our resting-state study conceptually thus resonates more with the concepts of the mesocircuit model, yet, we might hypothesize that relative preservation of the recurrent network crucial for the GNW meets the minimal requirement for conscious stimulus processing.

Our observation of altered spatio-temporal dynamics of functional connectivity that are also more restricted to the SC has been observed in pharmacological and pathological loss of consciousness (*Demertzi et al., 2019*; *López-González et al., 2021*; *Mashour et al., 2021*; *Uhrig et al., 2018*). During unconscious states it has been demonstrated that cerebral regional connectivity leads to an important reconfiguration of the repertoire of functional brain states, that forms simpler connectivity patterns in the brain, which is predominantly shaped by anatomical connectivity and is characterized by the reduction of complex long range connectivity patterns (*Barttfeld et al., 2014*; *Demertzi et al., 2019*; *Mashour et al., 2021*; *Uhrig et al., 2018*). Previous studies have already demonstrated a strong signature of the underlying structural network in functional networks in DoC. However, these previously described functional networks were mostly obtained using time-averaged estimation methods (*Annen et al., 2016*; *Enciso-Olivera et al., 2021*; *Kuceyeski et al., 2016*; *Lant et al., 2016*), and have not provided insight into a stronger structural functional coupling in DoC. Using time-resolved functional connectivity and its relationship to eigenmodes, we shed light on this observation. The underlying anatomical connectivity contains so-called inherent 'hidden patterns' or eigenmodes with different spatial structures. In a dynamical system such as the brain, these eigenmodes, or a combination of eigenmodes, can be sequentially activated or deactivated (*Atasoy et al., 2016*; *Tewarie et al., 2019*), and thereby shape the repertoire of possible functional network states. Since we assume that structural and functional networks share the same eigenmodes (*Abdelnour et al., 2018*; *Naze et al., 2021*; *Tewarie et al., 2020*), the weighting coefficients of the eigenmodes that result in the structural network can be considered as 'steady state'. Reconfigurations of spatiotemporal networks for the healthy brain are induced by strong deviations from this steady state, demonstrated by strong modulations in eigenmode expression in our work. In DoC however, there is strong tendency towards the 'steady state' with minimal deviations of eigenmode expressions over time, and hence leading to a strong signature of structural connections in functional networks in DoC. This inability was even more pronounced for UWS than for MCS.

A few methodological aspects of our retrospective study deserve further discussion. First of all, we did not analyse the contributions of individual eigenmodes for two reasons: (1) although earlier studies have demonstrated that a limited set of eigenmodes could already explain observed functional connectivity pattern (*Atasoy et al., 2016*), in our case, assessing group differences between MCS and UWS on the basis of individual eigenmodes would impose a multiple comparisons problem; (2) our analytical approach is based on the assumption that all eigenmodes are necessary in the mapping to functional connectivity instead of a statistical selection of eigenmodes. Second, concatenated data from all groups were fed into the NNTF analysis, instead of per group. This assumes that the spatial network structure is similar across groups. Even though this is not necessarily the case, the amount of data to allow for stable NNTF results for individual groups was limited, especially for the UWS group. Future (multicentric) studies with more patients should verify whether the decomposition of the dynamic functional connectivity patterns into the observed subnetworks holds for the separate groups. However, our approach to concatenate data from all groups made group comparison

much easier since there was now no need to 'match' potentially slightly different networks from the different groups. This also allowed us to focus on networks that were important in DoC. Lastly, we have used a high temporal resolution method to estimate time-resolved connectivity at every time point instead of a sliding window-based method. Previous studies using sliding window approaches have provided novel insights into brain dynamics of loss of consciousness, such as the occurrence and reconfiguration of brain states in DoC patients (*Demertzi et al., 2019*) and anaesthesia induced loss of consciousness (*Barttfeld et al., 2014*; *Uhrig et al., 2018*). However, sliding window approaches have limited sensitivity to nonstationarity in the fMRI BOLD signals (*Hindriks et al., 2016*). Recent work on time-resolved connectivity shows that brief periods in co-modulation of BOLD signals are an important driving factor for functional connectivity (*Zamani Esfahlani et al., 2020*; *Hindriks et al., 2016*), and sliding window approaches may be less sensitive to detect such brief periods in co-modulation. Furthermore, sliding window approaches in DoC have so far demonstrated limited success in retrieving canonical resting-state networks in DoC, while exploration of spatiotemporal aspects of well-known resting-state networks is important to better understand the relation between brain dynamics and loss of consciousness.

Taken together, we have demonstrated that a (semi) data-driven approach has extracted clinically meaningful time-resolved functional brain networks. This unique network-based spatiotemporal characterization accounts for structure–function coupling (i.e. eigenmodes), and shows a relationship with stability of brain dynamics. The measures that differed between UWS and MCS patients most, were the dominant eigenmodes (reflecting structure–function coupling) and time-resolved functional connectivity in the DMN, frontoparietal network, and the subcortical fronto-temporoparietal network. Interestingly, the latter network was generated by the (semi)data-driven approach to better fit the data, and was the sole network to show shorter dwell times in UWS compared to MCS patients. This suggests that the subcortical fronto-temporoparietal network might play a pivotal role in supporting conscious network interactions, as several theoretical and hypothesis-driven studies indicated. Future work will be required to assess to what extent these advanced aspects of connectivity can serve as biomarkers to aid diagnosis and prognosis in DoC.

## Methods

### Participants

Forty-four adult DoC patients, of whom 30 in MCS (11 females, age range 24–83 years; mean age ± SD, 45 ± 16 years) and 14 with the UWS (6 females, age range 20–74 years; mean age ± SD, 47 ± 16 years) and 34 age and gender matched HC subjects (14 females, age range 19–72 years; mean age ± SD, 40 ± 14 years) without premorbid neurological problems were included.

In this retrospective study, we used the same dataset as in *López-González et al., 2021*; *Panda et al., 2021*. The study was approved by the ethic committee of the the University Hospital of Liège (Belgium) (No. 2009-241), where all data were collected. Written informed consents to collect and published the data were obtained from all healthy subjects and the legal representative for DoC patients in accordance with the Declaration of Helsinki.

The diagnosis of the DoC patients was confirmed through two gold standard approaches that is (1) behavioural and (2) fluorodeoxyglucose-positron emission tomography ([18F]FDG-PET), excluding patients for whom these two diagnostic approaches disagreed. (1) Patients were behaviourally diagnosed through the best of at least five Coma Recovery Scale Revised (CRS-R) assessments, evaluating auditory, visual, motor, oromotor function, communication, and arousal (*Giacino et al., 2004*). (2) Behavioural diagnosis was complemented with the visual assessment of preserved brain metabolism in the frontoparietal network using [18F]FDG-PET as a neurological proxy for consciousness (*Stender et al., 2014*). Patient-specific clinical information is presented in *Supplementary file 1A*. We only included patients for whom (1) MRI data were recorded without anaesthesia, (2) diagnosis was based on at least five repetitions of the CRS-R assessment, (3) diagnosed as UWS or MCS, and (4) the [18F]FDG-PET diagnosis was in agreement with the clinical diagnosis. We excluded the patients (1) for whom the patients the structural MRI segmentation was incorrect or (2) if there were excessive head movement artefacts during MR recordings. There were 46 MCS patients in which 16 were discarded due to mismatch of PET and CRS-R diagnosis, 8 for failed segmentation and 4 for head movement

artefacts. Amongst the 28 UWS patients, 8 were discarded due to mismatch of the PET and CRS-R diagnosis, 4 for failure of segmentation, and 2 for head movement artefacts.

## MRI data acquisition

For the DoC dataset, structural (T1 and DWI) and functional MRI data were acquired using a Siemens 3T Trio scanner. 3D T1-weighted MP-RAGE images (120 transversal slices, repetition time = 2300 ms, voxel size = $1.0 \times 1.0 \times 1.2$ mm$^3$, flip angle = 9°, field of view = $256 \times 256$ mm$^2$) were acquired prior to the 10 min of BOLD fMRI resting-state (i.e. task free) (EPI, gradient echo, volumes = 300, repetition time = 2000 ms, echo time = 30 ms, flip angle = 78°, voxel size = $3 \times 3 \times 3$ mm$^3$, field of view = $192 \times 192$ mm$^2$, 32 transversal slices). HC subjects were instructed to keep eyes open and to be in relaxed state during the fMRI data acquisition. Last, diffusion-weighted MRI (DWI) was acquired in 64 directions (*b*-value = 1000 s/mm$^2$, voxel size = $1.8 \times 1.8 \times 3.3$ mm$^3$, field of view = $230 \times 230$ mm$^2$, repetition time = 5700 ms, echo time = 87 ms, 45 transverse slices, $128 \times 128$ voxel matrix) preceded by a single unweighted image (b0).

## Resting-state fMRI preprocessing

Preprocessing was performed as in *López-González et al., 2021* using MELODIC (Multivariate Exploratory Linear Optimized Decomposition into Independent Components) version 3.14, which is part of FMRIB's Software Library (FSL, http://fsl.fmrib.ox.ac.uk/fsl). The preprocessing consisted of the following steps: the first five functional images were discarded to reduce scanner inhomogeneity, motion correction was performed using MCFLIRT, non-brain tissue was removed using Bet Extraction Tool (BET), temporal band-pass filtering with sigma 100 s, spatial smoothing was applied using a 5-mm FWHM Gaussian kernel, rigid-body registration was performed, and finally single-session ICA with automatic dimensionality estimation was employed (*Griffanti et al., 2014*). Then, FIX (FMRIB's ICA-based X-noiseifier) was applied to remove the noise components and the lesion-driven for each subject. Specifically, FSLeyes in Melodic mode was used to manually identify the single-subject independent components (ICs) into 'good' for cerebral signal, 'bad' for noise or injury-driven artefacts, and 'unknown' for ambiguous components. Each component was evaluated based on the spatial map, the time series, and the temporal power spectrum (*Griffanti et al., 2014*). Next, for each subject, FIX was applied with default parameters to remove bad and unknown components. Subsequently, the Shen et al., functional atlas (without cerebellum) was applied for brain parcellation to obtain the BOLD time series of the 214 cortical and subcortical brain areas in each individual's native EPI space (*Finn et al., 2015*; *Shen et al., 2013*). The cleaned functional data were co-registered to the T1-weighted structural image by using FLIRT (*Jenkinson and Smith, 2001*). Then, the T1-weighted image was co-registered to the standard MNI space by using FLIRT (12 DOF) and FNIRT (*Andersson et al., 2007*). The transformations matrices were inverted and applied to warp the resting-state atlas from MNI space to the single-subject functional data using a nearest-neighbour interpolation method to ensure the preservation of the labels. Finally, the time series for each of the 214 brain areas were extracted using fslmaths and fslmeants.

## Probabilistic tractography analysis

A whole-brain SC matrix was computed for each subject as reported in our previous study (*López-González et al., 2021*). Briefly, the b0 image was co-registered to the T1 structural image using FLIRT and the T1 structural image was co-registered to the MNI space using FLIRT and FNIRT (*Andersson et al., 2007*; *Jenkinson and Smith, 2001*). The transformation matrices were inverted and applied to warp the resting-state atlas from MNI space to the native diffusion space using a nearest-neighbour interpolation method. Analysis of diffusion images was applied using the FMRIB's Diffusion Toolbox (FDT) (https://fsl.fmrib.ox.ac.uk/fsl/fslwiki). BET was computed, eddy current distortions and head motion were corrected using eddy correct tool (*Andersson and Sotiropoulos, 2016*). Crossing fibres were modelled by using BEDPOSTX, and the probability of multi-fibre orientations was calculated to enhance the sensitivity of non-dominant fibre populations (*Behrens et al., 2007*; *Behrens et al., 2003*). Then, probabilistic tractography analysis was calculated in native diffusion space using PROBTRACKX to compute the connectivity probability of each brain region to each of the other 213 brain regions. Subsequently, the value of each brain

area was divided by its corresponding number of generated tracts to obtain the structural probability matrix. Finally, the $SC_{pn}$ matrix was then symmetrized by computing their transpose $SC_{np}$ and averaging both matrices.

## Metastability and time-resolved functional connectivity

BOLD time series for every ROI $k$ were filtered within the narrowband of 0.01–0.9 Hz. We estimated both time-resolved functional connectivity and static functional connectivity. Static functional connectivity ($FC_{static}$) was estimated by the Pearson correlation coefficient between pairwise narrowband time courses. Time-resolved connectivity was estimated from the instantaneous phases. The instantaneous phase of the BOLD signal, $\varphi_k(t)$, was extracted from the analytic signal as obtained from the Hilbert transform. The synchronization between pairs of brain regions was characterized as the difference between their instantaneous phases. At each time point, the phase difference between two regions $j$ and $k$ was used as estimate for instantaneous phase connectivity $\Delta\varphi_{jk}(t)$ (**Glerean et al., 2012**). Evaluation of pairwise connectivity at each time point resulted in a functional connectivity tensor (number of ROIs × number of ROIs × time points).

Furthermore, we investigated how the synchronization between different nodes fluctuates across time using the concept of *metastability*. In the current sense, metastability quantifies how variable the states of phase configurations are as a function of time. We quantified metastability (i.e. our proxy measure for metastability) in terms of the standard deviation of the Kuramoto order parameter, $R(t) = \left| \frac{1}{N} \sum_k^N e^{i\varphi_k(t)} \right|$, where $N$ is the number of ROIs and $i$ denotes the imaginary unit (**Deco et al., 2017**).

Similar to **Ponce-Alvarez et al., 2015**; **Tewarie et al., 2019**, we extracted time-evolving networks using NNTF (**Bro, 1997**). We used the N-way toolbox (version 1.8) for MATLAB for this analysis (**Andersson and Bro, 2000**). NNTF can be considered as a higher-order principal component analysis. The goal of the approach is to decompose the functional connectivity tensor $FC$ into components, such that the approximation of the functional connectivity tensor $\widetilde{FC}$ can be written as $\widetilde{FC} = \sum_l^L a_l \times b_l \times c_l$. Here, $a_l$ and $b_l$ correspond to vectors decoding spatial information for component $l$, and $c_l$ represents the vector that contains information on temporal fluctuations of component $l$. Note that the outer product $a_l \times b_l$ stands for the spatial pattern of functional connectivity of component $l$. The number of components $L$ can be estimated from the data using established algorithms (**Timmerman and Kiers, 2000**). However, here, we fixed the number of components based on the number of a priori expected networks in which we were interested. We fed the NNTF algorithm with initial conditions for $a_l$ and $b_l$ based on the spatial components of six expected resting-state networks: salience network, FPN, DMN, subcortical network, sensorimotor network, and visual network. We added a residual network to account for the unexplained variance in the functional connectivity tensor. Note that the spatial initial conditions did not indicate that these spatial components were kept fixed during NNTF calculation; rather, these spatial components were free to be adjusted according to the maximization of the explained variance. Data from all subjects and groups were concatenated to allow convergence to stable results.

## Relationship between structural eigenmodes and time-resolved functional connectivity

For every subject, we extracted structural eigenmodes from the graph Laplacian $Q_A$ of the SC matrix defined by $Q_A = K_{SC} - SC$, where $K_{SC}$ refers to the diagonal degree matrix of SC. We further applied symmetric normalization to obtain a normalized Laplacian $Q_{SC}$. Subsequently, eigenvectors $u_i$ and eigenvalues (together called eigenmodes) were extracted using diagonalization of $Q_{SC}$, resulting in $N$ eigenmodes ($N$ = number of ROIs). Using a recently introduced approach we mapped functional brain networks at each time point from the structural eigenmodes (**Tewarie et al., 2022**; **Tewarie et al., 2020**). In other words, we estimated to what extent functional connectivity at each time point $FC(t)$ could be explained by a linear combination of the eigenmodes as follows:

$$FC(t) \approx K_{FC}(t) - K_{FC}^{\frac{1}{2}}(t) \left( UP(t) U^T \right) K_{FC}^{\frac{1}{2}}(t), \tag{1}$$

where $K_{FC}(t)$ is the diagonal node strength matrix of the functional connectivity matrix $FC(t)$ at time $t$. The matrix $P(t)$ corresponds to the weighting coefficient matrix for the eigenmodes and is obtained after optimization and equal to

$$P(t) = \text{diag}\left(u_1^T Q_{FC}(t) u_1, \ldots, u_N^T Q_{FC}(t) u_N\right) \tag{2}$$

Hence, modulations of eigenmode expressions over time are expressed in $P(t)$. Here, $Q_{FC}(t)$ is the normalized graph Laplacian of $FC(t)$ at time $t$, and $u_i$ is the $i$th eigenvector of $Q_{SC}$ and the $i$th column of $U$.

## Analysis steps

### Time-resolved functional connectivity

Individual temporal time courses $c_l^{ind}$ for expression of component $l$ for every subject were extracted from $c_l$. A high value of $c_l$ at a certain time point indicates strong expression of this spatial pattern of functional connectivity ($a_l \times b_l$) at that time point. At each time point, we determined the component with the strongest expression ($\max\left(c_l^{ind}\right)$), and assumed that connectivity at that point in time was dominated by this state or component. The duration that this component retained the strongest expression was considered as state duration or dwell time (see *Figure 1*). In addition, we also characterized the amount of nonstationarity in $c_l$ (*Hindriks et al., 2016*; *Zalesky et al., 2014*), that is excursions from the median. The rationale for using this metric is its sensitivity to detect modulations if the underlying system is indeed dynamic (*Hindriks et al., 2016*). Mann–Whitney $U$ tests were used to test, for each network separately, differences in state durations and excursion from the median between groups.

### Structural vs. functional brain networks

We first estimated the relationship between static functional connectivity and the SC itself (without decomposing SC into eigenmodes). This relationship was estimated in terms of the Pearson correlation coefficient between the SC and $FC_{static}$, denoted as $\text{corr}(FC_{static}, SC)$. We secondly analysed the extent to which time-varying functional networks could be explained by expressions of the structural eigenmodes. We therefore computed the Pearson correlation between the empirical $FC(t)$ and the eigenmode predicted $FC(t)$, denoted as $\text{corr}(FC, \text{eigenmodes})$. In order to be able to test whether there was a difference in fluctuations of eigenmode expression over time between groups, we quantified the eigenmode modulation strength defined as $\Delta\text{eigenmode} = \sum_{t_k=1, t_l=1}^{T,T} \| P(t_k) - P(t_j) \|$, where $t_j$ and $t_k$ correspond to different points in time and $T$ to the total duration of the recording. Since the first eigenmodes can be considered as dominant eigenmodes and more important to shape functional brain networks (*Atasoy et al., 2016*), we evaluated the eigenmode modulation strength for the dominant eigenmodes $i = \{1, \ldots, N/2\}$ and non-dominant eigenmodes $i = \{N/2, \ldots, N\}$ separately. We further computed the correlation between eigenmode modulation strength and metastability in all groups separately in order to test whether modulations in eigenmode expression related to our proxy measure for metastability. Group differences for all metrics was tested using Mann–Whitney $U$ tests.

In order to test whether eigenmode predictions of functional connectivity could be obtained by chance, we created surrogate data and redid analysis for the surrogate data. Surrogate data for fMRI BOLD time series were obtained using the circular time shifted method (*Quian Quiroga et al., 2002*). Time-resolved phase connectivity was estimated in the same way as for genuine empirical data. Time-resolved connectivity obtained from surrogate data was subsequently predicted using the eigenmodes.

False discovery rate was used to correct for multiple comparisons for analysis steps 1 and 2: number of metrics (excursions * seven networks, dwell time * seven networks, metastability, $\text{corr}(FC_{static}, SC)$, $\text{corr}(FC, \text{eigenmodes})$, non-dominant and dominant eigenmodes) * 3 comparisons + surrogate comparison with genuine data 3 * 3, ($\text{corr}(FC, \text{eigenmodes})$, non-dominant and dominant eigenmodes) comparisons = 66 tests (*Benjamini and Hochberg, 1995*).

# Additional information

## Funding

| Funder | Grant reference number | Author |
|---|---|---|
| Horizon 2020 Framework Programme | 945539 | Gustavo Deco Steven Laureys |
| Horizon 2020 Framework Programme | 785907 | Steven Laureys |
| Horizon 2020 Framework Programme | 778234 | Steven Laureys |
| The Belgian National Funds for Scientific Research (F.R.S-FNRS) | | Rajanikant Panda Aurore Thibaut Olivia Gosseries Steven Laureys |
| European Space Agency | | Steven Laureys |
| Swiss National Science Foundation | Sinergia (170873) | Ane Lopez-Gonzalez |
| Belgian Federal Science Policy Office | PRODEX Programme | Steven Laureys |
| Fondazione Europea di Ricerca Biomedica | | Steven Laureys |
| Bial Foundation | | Olivia Gosseries Steven Laureys |
| The fund Generet | | Steven Laureys |
| King Baudouin Foundation | | Steven Laureys |
| Leon Fredericq Foundation | | Steven Laureys Aurore Thibaut Olivia Gosseries |
| The DoCMA Project | EU-H2020-MSCA–RISE– 778234 | Aurore Thibaut |
| FLAG-ERA JTC | PCI2018-092891 | Gustavo Deco |
| The Spanish Ministry Project PSI2016-75688-P (AEI/FEDER) | | Gustavo Deco |
| The Catalan Research Group Support 2017 SGR 1 | | Gustavo Deco |
| Ministerio de Ciencia, Innovación y Universidades | | Gustavo Deco |
| Agencia Estatal de Investigación | | Gustavo Deco |
| European Regional Development Fund | | Gustavo Deco |
| Fonds De La Recherche Scientifique - FNRS | PDR project T.0134.21 | Aurore Thibaut Olivia Gosseries |
| Mind Science Foundation | | Steven Laureys |
| The Télévie Foundation | | Olivia Gosseries |
| The Public Utility Foundation 'Université Européenne du Travail' | | Steven Laureys |
| The Mind-Care foundation | | Steven Laureys |
| National Natural Science Foundation of China | Joint Research Project 81471100 | Steven Laureys |

| Funder | Grant reference number | Author |
|---|---|---|
| The European Foundation of Biomedical Research FERB Onlus | | Steven Laureys |
| The ERA-Net FLAG-ERA JTC2021 project ModelDXConsciousness (Human Brain Project Partnering Project) | | Olivia Gosseries |
| University and University Hospital of Liege | | Aurore Thibaut Olivia Gosseries Steven Laureys |

The funders had no role in study design, data collection, and interpretation, or the decision to submit the work for publication.

## Author contributions

Rajanikant Panda, Data curation, Formal analysis, Investigation, Visualization, Methodology, Writing – original draft, Writing – review and editing; Aurore Thibaut, Conceptualization, Data curation, Writing – review and editing, Methodology; Ane Lopez-Gonzalez, Anira Escrichs, Software, Formal analysis, Methodology, Writing – review and editing; Mohamed Ali Bahri, Validation, Methodology, Writing – review and editing; Arjan Hillebrand, Formal analysis, Supervision, Investigation, Visualization, Writing – review and editing; Gustavo Deco, Conceptualization, Supervision, Validation, Investigation, Methodology, Writing – review and editing; Steven Laureys, Conceptualization, Resources, Supervision, Funding acquisition, Investigation, Methodology, Writing – review and editing; Olivia Gosseries, Conceptualization, Resources, Data curation, Funding acquisition, Project administration, Writing – review and editing; Jitka Annen, Conceptualization, Resources, Formal analysis, Supervision, Investigation, Methodology, Writing – original draft, Writing – review and editing; Prejaas Tewarie, Conceptualization, Software, Formal analysis, Investigation, Visualization, Writing – original draft, Writing – review and editing

## Author ORCIDs

Rajanikant Panda ⓘ http://orcid.org/0000-0002-0960-4340
Anira Escrichs ⓘ http://orcid.org/0000-0002-6482-9737
Arjan Hillebrand ⓘ http://orcid.org/0000-0002-8508-3532
Prejaas Tewarie ⓘ http://orcid.org/0000-0002-3311-4990

## Ethics

Written informed consent was obtained from all healthy subjects and the legal representative for DoC patients.

## Decision letter and Author response

Decision letter https://doi.org/10.7554/eLife.77462.sa1
Author response https://doi.org/10.7554/eLife.77462.sa2

# Additional files

## Supplementary files

• Supplementary file 1. (A) Details about the patient population. (B) Brain regions involved in the extracted networks by non-negative tensor factorisation.

• MDAR checklist

## Data availability

Connectivity matrices have been made available open access through EBRAINS of the Human Brain Project.

The following previously published dataset was used:

| Author(s) | Year | Dataset title | Dataset URL | Database and Identifier |
|---|---|---|---|---|
| López-González A, Panda R, Barra A, Bonin EAC, Ponce-Alvarez A, Zamora-López G, Sanz LRD, Escrichs A, Cecconi B, Martial C, Thibaut A, Gosseries O, Annen J, Deco G, Laureys S | 2021 | Phase-interaction matrices of the phase relationships of different pathological states of consciousness | https://doi.org/10.25493/9GES-37K | Human Brain Project, 10.25493/9GES-37K |

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
