## [Editor Report]

This is an important paper providing convincing evidence for altered brain dynamics in patients in a minimally conscious state and those with unresponsive wakefulness syndrome relative to healthy control participants. The results indicate reduced metastability and a contracted network repertoire in disorders of consciousness. Overall, the study provides important new information on mechanisms of disorders of consciousness and the functional brain networks involved.

---

## [Decision Letter]

**Decision letter after peer review:**

Thank you for submitting your article "Disruption in structural-functional network repertoire and time-resolved subcortical-frontoparietal connectivity in disorders of consciousness" for consideration by *eLife*. Your article has been reviewed by 2 peer reviewers, and the evaluation has been overseen by a Reviewing Editor and Jonathan Roiser as the Senior Editor. The following individual involved in the review of your submission has agreed to reveal their identity: Patricio Orio (Reviewer #1).

Essential revisions:

1. There is frequent reference to "subcortical" and related networks, but I see no description in the text of which subcortical structures are involved. Panel N of Figure 2 is helpful but I think that more explicit detail is important, especially given the specific predictions of mesocircuit theory.

2. The authors have highlighted parallels with pharmacological states of unconsciousness and recovery therefrom. Two additional references to consider: Uhrig et al., (Anesthesiology, 2018, PMID: 30028727, which found similar convergence of structural/functional connectivity in the nonhuman primate neuronal workspace), and Mashour et al., (*eLife*, 2021, PMID: 33951427, which found support for global neuronal workspace in a human study of recovery from deep anaesthesia).

3. Similarly, although the global neuronal workspace does posit a critical role for recurrent frontal-parietal networks, can the authors be more specific about the nodes of the proposed workspace and what they found empirically?

4. The Results section gives the impression of a disconnected set of analyses that do not hang well together. In particular, the section about the structural eigenmodes and their relationship with the time-resolved FC seems to have little connection with the rest of the work, except for confirming (yet again) that DOC patients have a less dynamic FC. More elaboration about the relevance of these results, and what they say about DOC (that other dynamical FC analyses don't), is needed both in the Introduction and Discussion.

5. The classification sensitivity/specificity did not, in my opinion, add much to the manuscript, especially since the number of patients is not remotely close to what would be required for a population-based diagnostic approach. If the authors chose to include this with any reference to diagnosis (highlighted in the introduction and elsewhere), I would encourage a comparison with similar data from other clinical or neuroimaging-based diagnostic approaches. However, I think the value of the study resides more in terms of its mechanistic understanding than diagnosis.

6. Some comments about the connection(s) of these analyses with the commonly used FCD analysis (based on sliding windows of pair-wise correlations) will be useful, to put better this work into the big picture of time evolution of the functional connectivity.

7. The authors discuss how their data support the mesocircuit hypothesis and the global neuronal workspace hypothesis of conscious processing. However, these theories are neither interchangeable nor typically considered together. Could the authors be more specific in terms of how these theories interface vis-à-vis their empirical findings? Could they also be more explicit about whether their findings are inconsistent with other theories (e.g., IIT, predictive coding, higher-order thought)?

8. As a general point, the authors may have attempted too hard to give the paper a clinical/diagnostic (i.e., practical) emphasis; and in doing so, overlooked the clarification of some important issues. For example, clarifying the specific nodes under analysis and the relationship between the mesocircuit and global neuronal workspace theories is important. In addition, the authors need to consider whether this is primarily a neurobiological mechanisms study or a diagnostic/classification study. I would favour the former because the sensitivity/specificity analyses are not high yield and will require much more context in terms of other approaches.

*Reviewer #2 (Recommendations for the authors):*

1) The authors discuss how their data support the mesocircuit hypothesis and the global neuronal workspace hypothesis of conscious processing. However, these theories are neither interchangeable nor typically considered together. Could the authors be more specific in terms of how these theories interface vis-à-vis their empirical findings? Could they also be more explicit about whether their findings are inconsistent with other theories (e.g., IIT, predictive coding, higher-order thought)?

2) The authors have highlighted parallels with pharmacological states of unconsciousness and recovery therefrom. Two additional references to consider: Uhrig et al., (Anesthesiology, 2018, PMID: 30028727, which found similar convergence of structural/functional connectivity in the nonhuman primate neuronal workspace), and Mashour et al., (*eLife*, 2021, PMID: 33951427, which found support for global neuronal workspace in a human study of recovery from deep anesthesia).

The authors are to be commended for this important contribution to the field.

---

## [Author Response]

Essential revisions:1. There is frequent reference to "subcortical" and related networks, but I see no description in the text of which subcortical structures are involved. Panel N of Figure 2 is helpful but I think that more explicit detail is important, especially given the specific predictions of mesocircuit theory.

We have provided details for the subcortical networks presented in the Panel N of Figure 2. In the manuscript we provide a textual description of the brain areas that are part of the network. To improve the clarity of the description of the network, we also now refer to it as “subcortical fronto-temporoparietal (Sub-FTPN)”.

In the result section, it read as:

“This modulated subcortical fronto-temporoparietal network consist of the following brain regions: bilateral thalamus, caudate, right putamen, bilateral anterior and middle cingulate, inferior and middle frontal areas, supplementary motor cortex, middle and inferior temporal gyrus, right superior temporal, bilateral inferior parietal and supramarginal gyrus.”

2. The authors have highlighted parallels with pharmacological states of unconsciousness and recovery therefrom. Two additional references to consider: Uhrig et al., (Anesthesiology, 2018, PMID: 30028727, which found similar convergence of structural/functional connectivity in the nonhuman primate neuronal workspace), and Mashour et al., (eLife, 2021, PMID: 33951427, which found support for global neuronal workspace in a human study of recovery from deep anaesthesia).

We thank the reviewer for pointing us towards more relevant literature. We have now discussed and cited these two papers in the manuscript. The discussion on this topic now reads:

“Our observation of altered spatio-temporal dynamics of functional connectivity that are also more restricted to the structural connectivity has been observed in pharmacological and pathological loss of consciousness (Demertzi et al., 2019; López-González et al., 2021; Mashour et al., 2021; Uhrig et al., 2018). During unconscious states it has been demonstrated that cerebral regional connectivity leads to an important reconfiguration of the repertoire of functional brain states, that forms simpler connectivity patterns in the brain, which is predominantly shaped by anatomical connectivity and is characterized by the reduction of complex long range connectivity patterns (Barttfeld et al., 2014b; Demertzi et al., 2019; Mashour et al., 2021; Uhrig et al., 2018).”

3. Similarly, although the global neuronal workspace does posit a critical role for recurrent frontal-parietal networks, can the authors be more specific about the nodes of the proposed workspace and what they found empirically?

As above mentioned, we have provided more details about the regions part of the “subcortical fronto-temporoparietal”. As the reviewers rightfully noted, this network also shows some overlap with the Global Neuronal Workspace. We refer to that in more detail in the discussion, highlighting how our functional networks overlap and differ with the two networks (i.e., one feedforward only, one with recurrent activity), and with the predictions of the mesocircuit model. For more detail, please refer to our reply to point 7.

4. The Results section gives the impression of a disconnected set of analyses that do not hang well together. In particular, the section about the structural eigenmodes and their relationship with the time-resolved FC seems to have little connection with the rest of the work, except for confirming (yet again) that DOC patients have a less dynamic FC.

We understand the reviewer’s position. Part one of our work covers time-resolved FC and *spatiotemporal networks in* DoC. Part two covers the relationship between time-resolved FC and eigenmodes of the structural network. The rationale for including part two is the following: there is a lot of literature that shows that eigenmodes of the structural network can be considered as ‘building blocks’ or basis functions/vectors for spatiotemporal networks at the functional level (Aqil et al., 2021; Atasoy et al., 2016, 2018; Deslauriers-Gauthier et al., 2020; Gabay et al., 2018; Gabay and Robinson, 2017; Robinson et al., 2016; Robinson, 2021; Tewarie et al., 2019, 2020; Wang et al., 2017). Ideally to link part one and two, you would take this notion further by analysing if the magnitude eigenmode coefficients differed between UWS, MCS and healthy controls and how this would relate to dwell times or expression of *spatiotemporal networks*. However, this would lead to an immense multiple testing issue, which would be impossible to overcome with our sample size. An important link between part one and two of our work is the relationship between change in eigenmode expression and metastability. Our measure for metastability is only a proxy for metastability. Lack of change in eigenmode expressions seems to confirm this result of metastability.

To allow for better integration of part one and two of our work, we have added to the introduction:

“These eigenmodes can be considered as patterns of ‘hidden connectivity’ that come to expression at the level of functional networks. It has been postulated that eigenmodes form elementary building blocks for spatiotemporal dynamics (Aqil et al., 2021). There is evidence that the well-known resting state networks can be explained by activation of a small set of eigenmodes (Atasoy et al., 2018).”

We have also clarified in the result section:

“As resting-state network activity can be explained by activation of structural eigenmodes, we next analyse the role of fluctuations in eigenmode expression over time.”

More elaboration about the relevance of these results, and what they say about DOC (that other dynamical FC analyses don't), is needed both in the Introduction and Discussion.

We have now elaborated on the relevance of the eigenmode results in the discussion, which now reads:

“Previous studies have already demonstrated a strong signature of the underlying structural network in functional networks in DoC. However, these previously described functional networks were mostly obtained using *time-averaged* estimation methods (Annen et al., 2016; Enciso-Olivera et al., 2021; Kuceyeski et al., 2016; Lant et al., 2016), and have not provided insight into a stronger structural functional coupling in DoC. Using *time-resolved* functional connectivity and its relationship to eigenmodes, we shed light on this observation. The underlying anatomical connectivity contains so-called inherent ‘hidden patterns’ or eigenmodes with different spatial structures. In a dynamical system such as the brain, these eigenmodes, or a combination of eigenmodes, can be sequentially activated or deactivated (Atasoy et al., 2016; Tewarie et al., 2019), and thereby shape the repertoire of possible functional network states. Since we assume that structural and functional networks share the same eigenmodes (Abdelnour et al., 2018; Naze et al., 2021; Tewarie et al., 2020), the weighting coefficients of the eigenmodes that result in the structural network can be considered as ‘steady state’. Reconfigurations of spatiotemporal networks for the healthy brain are induced by strong deviations from this steady state, demonstrated by strong modulations in eigenmode expression in our work. In DoC however, there is strong tendency towards the ‘steady state’ with minimal deviations of eigenmode expressions over time, and hence leading to a strong signature of structural connections in functional networks in DoC. This was even more pronounced for UWS than for MCS.”

For clinical evaluation the structural support of the functional networks is less relevant as most variance is indeed explained by function, yet to understand the cerebral dysfunction a more coherent view is preferred. Moreover, as explained more in the response to point 5, we have removed the classification results from the manuscript to not deviate from our main findings.

5. The classification sensitivity/specificity did not, in my opinion, add much to the manuscript, especially since the number of patients is not remotely close to what would be required for a population-based diagnostic approach. If the authors chose to include this with any reference to diagnosis (highlighted in the introduction and elsewhere), I would encourage a comparison with similar data from other clinical or neuroimaging-based diagnostic approaches. However, I think the value of the study resides more in terms of its mechanistic understanding than diagnosis.

We agree with your suggestions that the primary aim of our work is to provide a mechanistic understanding of loss of consciousness. Therefore, we have removed the classification part from the paper and explain our findings focusing on mechanism of pathological unconsciousness rather than its potential as a clinical diagnostic tool. This change has required several textual edits throughout the manuscript.

6. Some comments about the connection(s) of these analyses with the commonly used FCD analysis (based on sliding windows of pair-wise correlations) will be useful, to put better this work into the big picture of time evolution of the functional connectivity.

We have now discussed sliding window-based analysis in the context of our work in the methodology section.

“Lastly, we have used a high temporal resolution method to estimate time-resolved connectivity at every time point instead of a sliding window-based method. Previous studies using sliding window approaches have provided novel insights into brain dynamics of loss of consciousness, such as the occurrence and reconfiguration of brain states in DoC patients (Demertzi et al., 2019) and anaesthesia induced loss of consciousness (Barttfeld et al., 2014a; Uhrig et al., 2018). However, sliding window approaches have limited sensitivity to non-stationarity in the fMRI BOLD signals (Hindriks et al., 2016). Recent work on time-resolved connectivity shows that brief periods in co-modulation of BOLD signals are an important driving factor for functional connectivity (Esfahlani et al., 2020; Hindriks et al., 2016), and sliding window approaches may be less sensitive to detect such brief periods in co-modulation. Furthermore, sliding window approaches in DoC have so far demonstrated limited success in retrieving canonical resting-state networks in DoC, while exploration of spatiotemporal aspects of well-known resting state networks is important to better understand the relation between brain dynamics and loss of consciousness.”

7. The authors discuss how their data support the mesocircuit hypothesis and the global neuronal workspace hypothesis of conscious processing. However, these theories are neither interchangeable nor typically considered together. Could the authors be more specific in terms of how these theories interface vis-à-vis their empirical findings? Could they also be more explicit about whether their findings are inconsistent with other theories (e.g., IIT, predictive coding, higher-order thought)?

Indeed, many other theories for the support of consciousness exist, and we do not discuss all. This is mostly because our study, from the methodological point of view, is limited to network function. Information integration, crucial for IIT, is not part of the study and linking our results to IIT would become too speculative. Likewise, we believe there are more appropriate experimental designs to study the predictive, that specifically allow to sensory input and error levels. The empirical predictions for the higher-order thought theory are not well developed, impairing the interpretation of our results in light to this theory.

The reviewer is correct that there is important overlap between the regions associated to the mesocircuit model and the GNW. We have therefore taken this opportunity to better clarify that in the discussion, which now reads:

“The mesocircuit hypothesis postulates that increased subcortical (specifically thalamic nuclei) inhibition and reduced excitatory output from the thalamus to the cortex (frontal, parietal, temporal, occipital) leads to weak neural activity and hampers consciousness in DoC patients (Fridman et al., 2014; Giacino et al., 2014; Schiff, 2010). Although our method does not allow to assess excitatory and inhibitory activity, it may be tempting to interpret that the subcortical fronto-temporoparietal network may play a crucial role in orchestrating global network interactions and dwell times.

Later in GNW paragraph, we added:

“Specifically, the GNW postulates that conscious perception should rely on two neural circuits. First, the feedforward circuit encompasses the visual and sensory areas and propagates sensory input to higher order integration areas. Second, in order for sensory input to be accessible to consciousness, recurrent activity in the circuit is required, responsible for ignition in a wide range of high-order functional areas including visual, parieto-temporal, thalamic and frontal areas (Aru et al., 2020; Baars, 2002; Dehaene and Changeux, 2011; Mashour et al., 2020). The subcortical fronto-temporoparietal network and frontoparietal network obtained in a data-driven fashion conjugates with the latter recurrent circuit proposed to be essential for consciousness by the GNW recurrent circuit. The breakdown of spatio-temporal coordination in the subcortical fronto-temporoparietal network and frontoparietal network was associated to reduced states of consciousness. So far, the importance of metastability has mainly been addressed in the context of recovery of consciousness from anaesthesia (Cavanna et al., 2018). Here, we underscore this finding and demonstrate that a reduction of subcortical-cortical spatiotemporal functional organization into metastable states can even differentiate between UWS and MCS patients aligning with key factors for consciousness as proposed by GNW and mesocircuit hypothesis.

Despite the overlapping cortical circuits important for the mesocircuit model and GNW, differences between the two hypotheses exist. While the occipital cortex is considered part of the same mesocircuit, its role is mostly for feedforward propagation in the GNW. We found a separate functional network that conceptually overlaps more with the feedforward network described by the GNW. In line with GNW predictions, the temporal dynamics in this feedforward network did not distinguish different states of consciousness (Dehaene and Changeux, 2011; Mashour et al., 2020). On the other hand, the involvement of subcortical areas in the subcortical fronto-temporoparietal network is specifically predicted by the mesocircuit model (Fridman et al., 2014; Giacino et al., 2014; Schiff, 2010). Last, the two theories highlight different aspects of brain activity required for consciousness. The GNW proposes specific effective connectivity patterns required for stimuli to reach consciousness. The mesocircuit model on the other hand proposes that balanced excitatory and inhibitory activity within the network is a prerequisite for conscious states to be supported. Our resting-state study conceptually thus resonates more with the concepts of the mesocircuit model, yet, we might hypothesize that relative preservation of the recurrent network crucial for the GNW meets the minimal requirement for conscious stimulus processing.”

8. As a general point, the authors may have attempted too hard to give the paper a clinical/diagnostic (i.e., practical) emphasis; and in doing so, overlooked the clarification of some important issues. For example, clarifying the specific nodes under analysis and the relationship between the mesocircuit and global neuronal workspace theories is important. In addition, the authors need to consider whether this is primarily a neurobiological mechanisms study or a diagnostic/classification study. I would favour the former because the sensitivity/specificity analyses are not high yield and will require much more context in terms of other approaches.

We agree with the reviewer. We have now focused the paper on mechanistic understanding of loss of consciousness rather than clinical/diagnostic impact.

We have modified the introduction and discussion in this direction in the current version of the paper. In the last paragraph of the Introduction, we explicitly state the aim of study:

“Finally, we conceptually link the findings of altered spatiotemporal dynamics underlying activity observed in the brain of DoC patients with several consciousness theories to increase our understanding of the mechanism behind pathological unconsciousness”.

Reviewer #2 (Recommendations for the authors):1) The authors discuss how their data support the mesocircuit hypothesis and the global neuronal workspace hypothesis of conscious processing. However, these theories are neither interchangeable nor typically considered together. Could the authors be more specific in terms of how these theories interface vis-à-vis their empirical findings? Could they also be more explicit about whether their findings are inconsistent with other theories (e.g., IIT, predictive coding, higher-order thought)?

Indeed, many other theories for the support of consciousness exist, and we do not discuss all. This is mostly because our study, from the methodological point of view, is limited to network function. Information integration, crucial for IIT, is not part of the study and linking our results to IIT would become too speculative. Likewise, we believe there are more appropriate experimental designs to study the predictive, that specifically allow to sensory input and error levels. The empirical predictions for the higher-order thought theory are not well developed, impairing the interpretation of our results in light to this theory.

The reviewer is correct that there is important overlap between the regions associated to the mesocircuit model and the GNW. We have therefore taken this opportunity to better clarify that in the discussion, which now reads:

“The mesocircuit hypothesis postulates that increased subcortical (specifically thalamic nuclei) inhibition and reduced excitatory output from the thalamus to the cortex (frontal, parietal, temporal, occipital) leads to weak neural activity and hampers consciousness in DoC patients (Fridman et al., 2014; Giacino et al., 2014; Schiff, 2010). Although our method does not allow to assess excitatory and inhibitory activity, it may be tempting to interpret that the subcortical fronto-temporoparietal network may play a crucial role in orchestrating global network interactions and dwell times.

Later in GNW paragraph, we added:

Specifically, the GNW postulates that conscious perception should rely on two neural circuits. First, the feedforward circuit encompasses the visual and sensory areas and propagates sensory input to higher order integration areas. Second, in order for sensory input to be accessible to consciousness, recurrent activity in the circuit is required, responsible for ignition in a wide range of high-order functional areas including visual, parieto-temporal, thalamic and frontal areas (Aru et al., 2020; Baars, 2002; Dehaene and Changeux, 2011; Mashour et al., 2020). The subcortical fronto-temporoparietal network and frontoparietal network obtained in a data-driven fashion conjugates with the latter recurrent circuit proposed to be essential for consciousness by the GNW recurrent circuit. The breakdown of spatio-temporal coordination in the subcortical fronto-temporoparietal network and frontoparietal network was associated to reduced states of consciousness. So far, the importance of metastability has mainly been addressed in the context of recovery of consciousness from anaesthesia (Cavanna et al., 2018). Here, we underscore this finding and demonstrate that a reduction of subcortical-cortical spatiotemporal functional organization into metastable states can even differentiate between UWS and MCS patients aligning with key factors for consciousness as proposed by GNW and mesocircuit hypothesis.

Despite the overlapping cortical circuits important for the mesocircuit model and GNW, differences between the two hypotheses exist. While the occipital cortex is considered part of the same mesocircuit, its role is mostly for feedforward propagation in the GNW. We found a separate functional network that conceptually overlaps more with the feedforward network described by the GNW. In line with GNW predictions, the temporal dynamics in this feedforward network did not distinguish different states of consciousness (Dehaene and Changeux, 2011; Mashour et al., 2020). On the other hand, the involvement of subcortical areas in the subcortical fronto-temporoparietal network is specifically predicted by the mesocircuit model (Fridman et al., 2014; Giacino et al., 2014; Schiff, 2010). Last, the two theories highlight different aspects of brain activity required for consciousness. The GNW proposes specific effective connectivity patterns required for stimuli to reach consciousness. The mesocircuit model on the other hand proposes that balanced excitatory and inhibitory activity within the network is a prerequisite for conscious states to be supported. Our resting-state study conceptually thus resonates more with the concepts of the mesocircuit model, yet, we might hypothesize that relative preservation of the recurrent network crucial for the GNW meets the minimal requirement for conscious stimulus processing.”

2) The authors have highlighted parallels with pharmacological states of unconsciousness and recovery therefrom. Two additional references to consider: Uhrig et al., (Anesthesiology, 2018, PMID: 30028727, which found similar convergence of structural/functional connectivity in the nonhuman primate neuronal workspace), and Mashour et al., (eLife, 2021, PMID: 33951427, which found support for global neuronal workspace in a human study of recovery from deep anesthesia).

We thank the reviewer for pointing us towards more relevant literature. We have now discussed and cited these two papers in the manuscript. The discussion on this topic now reads:

“Our observation of altered spatio-temporal dynamics of functional connectivity that are also more restricted to the structural connectivity has been observed in pharmacological and pathological loss of consciousness (Demertzi et al., 2019; López-González et al., 2021; Mashour et al., 2021; Uhrig et al., 2018). During unconscious states it has been demonstrated that cerebral regional connectivity leads to an important reconfiguration of the repertoire of functional brain states, that forms simpler connectivity patterns in the brain, which is predominantly shaped by anatomical connectivity and is characterized by the reduction of complex long range connectivity patterns (Barttfeld et al., 2014b; Demertzi et al., 2019; Mashour et al., 2021; Uhrig et al., 2018).”

References

Abdelnour F, Dayan M, Devinsky O, Thesen T, Raj A. 2018. Functional brain connectivity is predictable from anatomic network’s Laplacian eigen-structure. *Neuroimage* 172. doi:10.1016/j.neuroimage.2018.02.016

Annen J, Heine L, Ziegler E, Frasso G, Bahri M, Di Perri C, Stender J, Martial C, Wannez S, D’ostilio K. 2016. Function–structure connectivity in patients with severe brain injury as measured by MRI‐DWI and FDG‐PET. *Hum Brain Mapp* 37:3707–3720.

Aqil M, Atasoy S, Kringelbach ML, Hindriks R. 2021. Graph neural fields: A framework for spatiotemporal dynamical models on the human connectome. *PLoS Comput Biol* 17. doi:10.1371/JOURNAL.PCBI.1008310

Aru J, Suzuki M, Larkum ME. 2020. Cellular Mechanisms of Conscious Processing. *Trends Cogn Sci*. doi:10.1016/j.tics.2020.07.006

Atasoy S, Deco G, Kringelbach ML, Pearson J. 2018. Harmonic Brain Modes: A Unifying Framework for Linking Space and Time in Brain Dynamics. *Neuroscientist*. doi:10.1177/1073858417728032

Atasoy S, Donnelly I, Pearson J. 2016. Human brain networks function in connectome-specific harmonic waves. *Nat Commun* 7. doi:10.1038/ncomms10340

Baars BJ. 2002. The conscious access hypothesis: Origins and recent evidence. *Trends Cogn Sci*. doi:10.1016/S1364-6613(00)01819-2

Barttfeld P, Uhrig L, Sitt JD, Sigman M, Jarraya B, Dehaene S. 2014a. Signature of consciousness in the dynamics of resting-state brain activity. *Proc Natl Acad Sci* 112:201418031.

Barttfeld P, Uhrig L, Sitt JD, Sigman M, Jarraya B, Dehaene S. 2014b. Signature of consciousness in the dynamics of resting-state brain activity. *Proc Natl Acad Sci* 112:201418031. doi:10.1073/pnas.1418031112

Cavanna F, Vilas MG, Palmucci M, Tagliazucchi E. 2018. Dynamic functional connectivity and brain metastability during altered states of consciousness. *Neuroimage* 180:383–395.

Dehaene S, Changeux JP. 2011. Experimental and Theoretical Approaches to Conscious Processing. *Neuron*. doi:10.1016/j.neuron.2011.03.018

Demertzi A, Tagliazucchi E, Dehaene S, Deco G, Barttfeld P, Raimondo F, Martial C, Fernández-Espejo D, Rohaut B, Voss HU, Schiff ND, Owen AM, Laureys S, Naccache L, Sitt JD. 2019. Human consciousness is supported by dynamic complex patterns of brain signal coordination. *Sci Adv* 5. doi:10.1126/sciadv.aat7603

Deslauriers-Gauthier S, Zucchelli M, Frigo M, Deriche R. 2020. A unified framework for multimodal structure–function mapping based on eigenmodes. *Med Image Anal* 66. doi:10.1016/j.media.2020.101799

Enciso-Olivera CO, Ordóñez-Rubiano EG, Casanova-Libreros R, Rivera D, Zarate-Ardila CJ, Rudas J, Pulido C, Gómez F, Martínez D, Guerrero N, Hurtado MA, Aguilera-Bustos N, Hernández-Torres CP, Hernandez J, Marín-Muñoz JH. 2021. Structural and functional connectivity of the ascending arousal network for prediction of outcome in patients with acute disorders of consciousness. *Sci Rep* 11. doi:10.1038/s41598-021-98506-7

Esfahlani FZ, Jo Y, Faskowitz J, Byrge L, Kennedy DP, Sporns O, Betzel RF. 2020. High-amplitude cofluctuations in cortical activity drive functional connectivity. *Proc Natl Acad Sci U S A* 117. doi:10.1073/pnas.2005531117

Fridman EA, Beattie BJ, Broft A, Laureys S, Schiff ND. 2014. Regional cerebral metabolic patterns demonstrate the role of anterior forebrain mesocircuit dysfunction in the severely injured brain. *Proc Natl Acad Sci* 111. doi:10.1073/pnas.1320969111

Gabay NC, Babaie-Janvier T, Robinson PA. 2018. Dynamics of cortical activity eigenmodes including standing, traveling, and rotating waves. *Phys Rev E* 98. doi:10.1103/PhysRevE.98.042413

Gabay NC, Robinson PA. 2017. Cortical geometry as a determinant of brain activity eigenmodes: Neural field analysis. *Phys Rev E* 96. doi:10.1103/PhysRevE.96.032413

Giacino JT, Fins JJ, Laureys S, Schiff ND. 2014. Disorders of consciousness after acquired brain injury: the state of the science. *Nat Rev Neurol* 10:99.

Hindriks R, Adhikari MH, Murayama Y, Ganzetti M, Mantini D, Logothetis NK, Deco G. 2016. Can sliding-window correlations reveal dynamic functional connectivity in resting-state fMRI? *Neuroimage* 127:242–256.

Kuceyeski A, Shah S, Dyke JP, Bickel S, Abdelnour F, Schiff ND, Voss HU, Raj A. 2016. The application of a mathematical model linking structural and functional connectomes in severe brain injury. *NeuroImage Clin* 11. doi:10.1016/j.nicl.2016.04.006

Lant ND, Gonzalez-Lara LE, Owen AM, Fernández-Espejo D. 2016. Relationship between the anterior forebrain mesocircuit and the default mode network in the structural bases of disorders of consciousness. *NeuroImage Clin* 10. doi:10.1016/j.nicl.2015.11.004

López-González A, Panda R, Ponce-Alvarez A, Zamora-López G, Escrichs A, Martial C, Thibaut A, Gosseries O, Kringelbach ML, Annen J, Laureys S, Deco G. 2021. Loss of consciousness reduces the stability of brain hubs and the heterogeneity of brain dynamics. *Commun Biol* 4:2020.11.20.391482. doi:10.1038/s42003-021-02537-9

Mashour GA, Palanca BJA, Basner M, Li D, Wang W, Blain-Moraes S, Lin N, Maier K, Muench M, Tarnal V, Vanini G, Ochroch EA, Hogg R, Schwartz M, Maybrier H, Hardie R, Janke E, Golmirzaie G, Picton P, McKinstry-Wu AR, Avidan MS, Kelz MB. 2021. Recovery of consciousness and cognition after general anesthesia in humans. *eLife* 10. doi:10.7554/*eLife*.59525

Mashour GA, Roelfsema P, Changeux JP, Dehaene S. 2020. Conscious Processing and the Global Neuronal Workspace Hypothesis. *Neuron*. doi:10.1016/j.neuron.2020.01.026

Naze S, Proix T, Atasoy S, Kozloski JR. 2021. Robustness of connectome harmonics to local gray matter and long-range white matter connectivity changes: Sensitivity analysis of Connectome Harmonics. *Neuroimage* 224. doi:10.1016/j.neuroimage.2020.117364

Robinson PA. 2021. Discrete spectral eigenmode-resonance network of brain dynamics and connectivity. *Phys Rev E* 104. doi:10.1103/PhysRevE.104.034411

Robinson PA, Zhao X, Aquino KM, Griffiths JD, Sarkar S, Mehta-Pandejee G. 2016. Eigenmodes of brain activity: Neural field theory predictions and comparison with experiment. *Neuroimage* 142:79–98.

Schiff ND. 2010. Recovery of consciousness after brain injury: a mesocircuit hypothesis. *Trends Neurosci*. doi:10.1016/j.tins.2009.11.002

Tewarie P, Abeysuriya R, Byrne Á, O’Neill GC, Sotiropoulos SN, Brookes MJ, Coombes S. 2019. How do spatially distinct frequency specific MEG networks emerge from one underlying structural connectome? The role of the structural eigenmodes. *Neuroimage* 186. doi:10.1016/j.neuroimage.2018.10.079

Tewarie P, Prasse B, Meier JM, Santos FAN, Douw L, Schoonheim MM, Stam CJ, Van Mieghem P, Hillebrand A. 2020. Mapping functional brain networks from the structural connectome: Relating the series expansion and eigenmode approaches. *Neuroimage* 216. doi:10.1016/j.neuroimage.2020.116805

Uhrig L, Sitt JD, Jacob A, Tasserie J, Barttfeld P, Dupont M, Dehaene S, Jarraya B. 2018. Resting-state Dynamics as a Cortical Signature of Anesthesia in Monkeys. *Anesthesiology* 129. doi:10.1097/ALN.0000000000002336

Wang MB, Owen JP, Mukherjee P, Raj A. 2017. Brain network eigenmodes provide a robust and compact representation of the structural connectome in health and disease. *PLoS Comput Biol* 13. doi:10.1371/journal.pcbi.1005550